# Pregnancy Care in Times of Cannabis Legalization: Self-Rated Knowledge, Risk Perception and Communication Practices of Midwives in Germany

**DOI:** 10.3390/healthcare13111228

**Published:** 2025-05-23

**Authors:** Julia Wollscheid, Matthias Burke, Theresa Kimmel, Tobias Kaufmann, Anil Batra, Annette Binder

**Affiliations:** 1Institute for Health Sciences, Department of Midwifery Science, University Hospital Tübingen, 72076 Tübingen, Germany; julia.wollscheid@student.uni-tuebingen.de; 2Department of General Psychiatry and Psychotherapy, Addiction Medicine and Addiction Research Section, University Hospital Tübingen, 72076 Tübingen, Germany; matthias.burke@student.uni-tuebingen.de (M.B.); theresa-marlen.kimmel@student.uni-tuebingen.de (T.K.); anil.batra@med.uni-tuebingen.de (A.B.); 3Department of Psychiatry and Psychotherapy, Tübingen Center for Mental Health, University of Tübingen, 72076 Tübingen, Germany; tobias.kaufmann@med.uni-tuebingen.de; 4German Center for Mental Health (DZPG), Partner Site Tübingen, 72076 Tübingen, Germany; 5Centre for Precision Psychiatry, Division of Mental Health and Addiction, Institute of Clinical Medicine, University of Oslo, 0424 Oslo, Norway

**Keywords:** prenatal cannabis use, midwifery, risk perception, maternal health, pregnancy, counseling, professional education, risk perception, legalization, Germany

## Abstract

Background/Objectives: The legalization of recreational cannabis in Germany in 2024 has increased the relevance of cannabis use in maternal healthcare. Although prenatal cannabis exposure is associated with potential risks to fetal development, the topic remains underrepresented in midwifery research and education. Germany, midwives play a key role in prenatal care. This study investigates midwives’ self-rated knowledge, perceived risks, and the frequency of screening and counseling on cannabis use during pregnancy. Methods: This study presents a secondary analysis of cross-sectional survey data collected from midwives and physicians in Germany (N = 284) between May and October 2024. Statistical analyses included descriptive statistics, Wilcoxon signed-rank tests, chi-square tests, Spearman’s rank correlations, and multiple linear regression models. Results: Midwives rated their knowledge about cannabis-related risks during pregnancy as moderate. While most reported that substance use was addressed in initial training, only continuing education and older age were associated with higher self-rated knowledge. Knowledge was positively correlated with risk perception and communication frequency. Overall, risk perception was high—particularly regarding fetal outcomes—though cannabis was perceived as less harmful than alcohol and addressed less often. Fewer than half of the midwives routinely screened for cannabis use, and only 22% always provided counseling. Conclusions: To strengthen midwives’ preparedness, both the integration of cannabis-specific content into initial training and the expansion of continuing education may be beneficial. Clear, evidence-based, and non-stigmatizing communication strategies are essential to support maternal and fetal health in a changing legal and cultural landscape.

## 1. Introduction

In recent years, social acceptance of cannabis use has increased, while legal regulation has become more liberal in many countries, including Germany. Following a global trend toward decriminalization and legalization, Germany legalized recreational cannabis for adults in April 2024 [1]. This development has made the substance more accessible to a broader population, including individuals of reproductive age, and raises important questions for maternal and fetal health.

Although cannabis is often perceived as a “natural” or safer alternative to other substances, studies suggest that prenatal cannabis use may have adverse effects. These include impaired placental function [2,3,4], increased risk of intrauterine growth restriction (IUGR) [5,6,7], low birth weight (LBW) [8], and neurodevelopmental disorders in the offspring [9,10,11,12]. The underlying mechanisms involve a disruption of the endocannabinoid system, which plays a crucial role in placental development and fetal brain maturation. Δ9-Tetrahydrocannabinol (THC), the main psychoactive component of cannabis, can cross the placenta, bind to cannabinoid receptor type 1 (CB1), and impair trophoblast function, placental perfusion, as well as the fetal supply of oxygen and nutrients [2,3,4].

Despite these concerns, the scientific evidence remains inconclusive. Risk assessment is complicated by methodological challenges such as co-use of tobacco and other substances, variations in dosage and timing of use, socio-demographic factors, and the reliance on self-reported data [10,13]. While some studies have statistically controlled for these confounders and found that cannabis alone does not significantly increase the risk for LBW, preterm birth, or being small for gestational age (SGA) [13], behavioral effects such as increased aggression or attention problems have persisted even after adjustment for maternal age, socio-economic status, and additional substance use [11]. These findings are further supported by animal studies, which demonstrate marked behavioral alterations in rodents following prenatal THC exposure [14,15].

International prevalence rates of prenatal cannabis use vary widely. In the United States, estimates range from 2% to 27%, depending on the population studied and whether self-reports or toxicological screenings are used [16,17]. In France, around 1.1% of pregnant individuals report using cannabis during pregnancy [18]. Germany lacks national data on prenatal cannabis use; however, general consumption trends suggest growing relevance. According to the most recent data, lifetime cannabis use among 18- to 25-year-olds exceeds 50%, and the 12-month prevalence was projected to reach 12.7% in 2024 [19,20]. Given these trends, it is plausible that a proportion of pregnant individuals in Germany continue to use cannabis.

Motivations for cannabis use during pregnancy are complex. Many individuals report using cannabis to manage symptoms such as nausea, vomiting, anxiety, or pain—particularly when conventional treatments are perceived as ineffective [21,22,23,24,25]. Others consider cannabis to be a less harmful alternative to alcohol, tobacco, or prescription medication [26]. Notably, research suggests that many pregnant individuals critically reflect on their consumption and adopt harm-reduction strategies, such as lowering the dose, changing the method of consumption, or abstaining during certain stages of pregnancy, often with the fetus’s well-being in mind [27].

In this context, clear, evidence-based guidance is essential. In Germany, antenatal care can be provided either by gynecologists or by midwives. While most pregnant individuals opt for gynecologist-led care, midwives can provide all routine check-ups except for the three major ultrasound scans, which must be performed by a physician. Midwifery-led antenatal visits often include not only clinical assessments but also conversations about emotional well-being, stress, or social concerns [28]. This creates a particularly safe and trusting environment—one in which pregnant individuals may feel more comfortable disclosing cannabis use than in a clinical setting. For this reason, both midwives and gynecologists are uniquely positioned to identify cannabis use, initiate open conversations, and offer individualized, non-judgmental support throughout pregnancy.

However, several studies have highlighted considerable knowledge gaps, a lack of sufficient training, and the absence of standardized guidelines for addressing substance use in pregnancy. As a result, many professionals report uncertainty or avoidance, relying on personal attitudes rather than scientific evidence [29,30,31]. Risk perception among health professionals also varies widely and is influenced not only by clinical knowledge but also by legal, cultural, and personal beliefs. While some perceive cannabis use as highly risky—especially in terms of fetal outcomes—others consider it less harmful than alcohol or tobacco [30]. Consequently, counseling practices range from abstinence-oriented or punitive messages to harm-reduction approaches. This inconsistency can undermine trust, discourage disclosure, and compromise the therapeutic relationship [31].

Despite its growing relevance—especially in light of legalization—research on how midwives in Germany perceive and manage prenatal cannabis use remains scarce. This is particularly relevant in the context of recent legalization, which may affect not only usage patterns but also professional attitudes and practices. Compared to alcohol, which is well-integrated into clinical guidelines and professional education, cannabis appears to be a regulatory and educational blind spot. Moreover, there is limited comparative research exploring how professionals assess and communicate about cannabis relative to other substances [32].

This study aims to investigate midwives’ knowledge, risk perception, and communication practices regarding cannabis use during pregnancy in Germany. It also examines the potential influence of age, professional background, and training on these factors. Selected comparisons with physicians are included to contextualize findings across professional groups. By identifying current challenges and needs, this research seeks to inform future educational efforts and support the development of clear, evidence-based, and non-stigmatizing counseling strategies in midwifery care.

## 2. Materials and Methods

### 2.1. Study Design and Sample

This study presents a secondary, exploratory analysis of data collected as part of the project “Understanding, Supporting, Motivating: Together for an Alcohol-Free Pregnancy”, developed by the Department of Addiction Medicine and Addiction Research at the University Hospital of Psychiatry and Psychotherapy Tübingen. Funded by the Federal Institute of Public Health (BIÖG), the project aimed to strengthen healthcare professionals’ competencies in addressing substance use during pregnancy through the development and evaluation of a web-based e-learning program.

As part of the project, a cross-sectional online survey was conducted independently between 6 May and 2 October 2024. This survey served two purposes: it provided baseline data for the evaluation of the e-learning program and was also open to a broader sample beyond e-learning participants.

### 2.2. Study Population and Recruitment

The original target group included healthcare professionals involved in antenatal care, such as midwives, physicians, and staff from pregnancy and family counseling services. For this secondary analysis, only responses from midwives and, where relevant, physicians were included, as these two groups are primarily responsible for providing medical antenatal care in Germany.

Participants were required to have sufficient German language skills and internet access. Recruitment was carried out through multiple channels, including emails and faxes to midwifery practices, birth centers, and professional associations such as the German Midwives Association. In addition, the survey was distributed via social media and snowball sampling.

Due to the open and decentralized recruitment strategy, the exact number of professionals who received the survey invitation could not be determined. Participation was voluntary. As an incentive, participants were given the opportunity to enter a raffle to win one of five €100 vouchers and to register for free access to the developed e-learning program.

### 2.3. Questionnaire and Data Collection

The questionnaire was developed by the project’s research team and included 42–48 items, depending on branching logic. It assessed sociodemographic characteristics, self-reported substance use, knowledge regarding substance use in pregnancy, risk perception, and communication practices. A full overview of the included variables, their scales, and statistical tests is presented in Table 1; the complete questionnaire is available in Appendix B.

A pilot test with eight individuals—including two gynecologists, two midwives, two social workers, and two laypersons—was conducted to evaluate clarity and usability. Minor revisions were made based on their feedback. The final version was administered via LimeSurvey (6.3.8 + 231,204) and took approximately 15–20 min to complete.

Participants accessed the survey via a secure LimeSurvey link. Informed consent was obtained digitally via checkbox prior to participation. Email addresses provided for the raffle or training access were stored separately from survey responses to ensure anonymity.

### 2.4. Data Preparation and Analysis

The dataset (N = 639) was exported to R, version 4.4.1 for initial cleaning. In line with the data-cleaning criteria proposed by Schendera [33], incomplete responses (n = 172) and duplicates (n = 15), identified via matching sociodemographic data, were removed. Only the first complete entry was retained in cases of duplication.

For statistical analysis, IBM SPSS Statistics Version 29.0 (IBM Corp., Armonk, NY, USA) was used. Most analyses were restricted to midwives (n = 143). For selected comparisons, data from physicians (n = 141) were also included, resulting in a final sample of N = 284.

Descriptive statistics were used to summarize sample characteristics and key study variables, including means (M) and standard deviations (SD) for continuous data and frequencies and percentages for categorical data.

To examine group differences between midwives and physicians, independent-samples *t*-tests were used for approximately normally distributed continuous variables. For ordinal variables such as screening and counseling frequencies, chi-square (χ^2^) tests were applied.

Wilcoxon signed-rank tests (Z) were conducted for within-subject comparisons of non-normally distributed ordinal variables, such as perceived risks of cannabis versus alcohol use during pregnancy.

Associations between continuous or ordinal variables (e.g., age, knowledge level, and risk perception) were analyzed using Spearman’s rank correlation coefficients (ρ). Correlation coefficients were interpreted according to Cohen’s guidelines: small (ρ = 0.1), medium (ρ = 0.3), and large (ρ = 0.5). For group comparisons, effect sizes were reported using Cohen’s d (small = 0.2, medium = 0.5, large = 0.8).

Multiple linear regression analyses were used to examine the influence of age and knowledge level on outcomes such as risk perception and communication behavior. The results are reported as standardized regression coefficients (β), along with the F-statistics of the model. Prior to regression analyses, relevant assumptions were examined. Due to strong collinearity between age and years of professional experience (ρ = 0.94, *p* < 0.001), only age was included as a continuous predictor in the regression models. Age was chosen because generational effects and cultural influences are expected to shape professionals’ knowledge and attitudes more strongly than years of practice.

Given the exploratory nature of the study, inferential statistical tests were conducted to identify potential patterns and associations but should be interpreted cautiously. The analyses were not based on predefined hypotheses and primarily serve to generate hypotheses for future research.

All statistical tests were two-tailed, and a *p*-value < 0.05 was considered statistically significant [34].

### 2.5. Ethical and Data Protection Aspects

The original study was approved by the Ethics Committee of the Medical Faculty, University of Tübingen (Ref. 067/2024BO1). Participants were informed about the study purpose, data protection measures, and their rights. Informed consent was obtained digitally via a checkbox before survey participation.

LimeSurvey, the survey platform, complies with the European Union General Data Protection Regulation (EU GDPR). Data were stored on ISO 27001-certified servers in Germany, encrypted via SSL protocols [35].

No identifying information was collected. Identifiable information, such as email addresses for incentives or training registration, was stored separately and not linked to survey responses. Data were transferred to university servers post-collection and analyzed by authorized team members. Data will be archived securely for at least 10 years.

## 3. Results

### 3.1. Sociodemographic Characteristics of the Sample

A total of 284 participants were included in the analysis, comprising 143 midwives and 141 physicians. Table 2 presents the sociodemographic and professional characteristics of the study sample, as well as additional variables such as previous training on substance use and personal cannabis consumption. On average, midwives were 38.8 years old and had 14.3 years of professional experience. The vast majority reported being biologically female and also identified as female in terms of their gender. Most midwives worked in non-hospital settings (54,5%) such as birth centers or midwifery practices. Compared to physicians, midwives were significantly younger (t(269) = 5.29, *p* < 0.001) and had fewer years of professional experience (t(282) = 3.25, *p* = 0.001). Age and years of experience were highly correlated (ρ = 0.94, *p* < 0.001). A comprehensive overview of all statistical tests, group differences, and effect sizes can be found in Appendix A.

### 3.2. Midwives’ Knowledge About Cannabis Use During Pregnancy

#### 3.2.1. Self-Rated Knowledge Level

On a scale from 0 (“no knowledge”) to 10 (“very high knowledge”), midwives rated their knowledge about cannabis use during pregnancy at an average of 4.71 (moderate knowledge). The large standard deviation (SD = 2.93) and wide distribution of responses (see Figure 1) suggest substantial interindividual variation. Specifically, 21.7% of participants rated their knowledge as high (8–10), while 41.3% rated it in the lower range (0–3).

Compared to their self-rated knowledge regarding cannabis, participants rated their knowledge about alcohol during pregnancy significantly higher (M = 6.93, SD = 1.79). A paired-samples *t*-test confirmed this difference to be statistically significant, t(142) = −9.09, *p* < 0.001, with a large effect size according to Cohen’s classification (d = 0.76). A moderate positive correlation was also found between the two ratings, r = 0.31, *p* < 0.001, suggesting that participants who felt more knowledgeable about alcohol use also tended to feel more knowledgeable about cannabis use.

Notably, the majority of midwives (83.9%) expressed a high or very high need for further training on cannabis use in pregnancy.

#### 3.2.2. Education and Training

Among midwives, 78.3% reported that the topic of substance use during pregnancy had been addressed during their initial professional training. Furthermore, 35.7% of midwives indicated that they had attended a continuing education course on this topic.

Physicians reported slightly lower rates for both categories, with 65.2% stating that the topic had been included in their professional training and 34.8% having attended further training. A chi-square test revealed that only the difference in initial training between the two groups was statistically significant (χ^2^(1) = 5.997, *p* = 0.014). No significant difference was found regarding continuing education attendance (χ^2^(1) = 0.026, *p* = 0.872).

Despite differences in initial training, midwives and physicians reported similar perceived knowledge regarding cannabis use in pregnancy (M = 4.71, SD = 2.93 vs. M = 5.20, SD = 2.43). The difference was not statistically significant (t(274) = 1.540, *p* = 0.125), and the corresponding effect size was small (Cohen’s d = 0.18).

To assess whether training and continuing education were associated with perceived knowledge about cannabis use during pregnancy, separate t-tests were conducted. No significant difference was found between those midwives who received training as part of their education (M = 4.60, SD = 2.95) and those who did not (M = 5.10, SD = 2.87), t(141) = −0.851, *p* = 0.399, d = −0.17.

In contrast, participation in continuing education was associated with significantly higher self-rated knowledge: midwives who had attended such training reported higher knowledge levels (M = 5.59, SD = 2.80) than those who had not (M = 4.22, SD = 2.90), t(106.6) = 2.77, *p* = 0.007, d = 0.48.

#### 3.2.3. Influence of Age

A moderate positive correlation was found between age and self-rated knowledge (ρ = 0.39, *p* < 0.001), suggesting that older midwives felt more confident in their knowledge about cannabis use during pregnancy compared to younger colleagues.

### 3.3. Risk Perception of Cannabis Use During Pregnancy

#### 3.3.1. Perceived Risks of Cannabis Use for the Fetus and the Pregnant Person

Midwives rated the risk of prenatal cannabis use as high for both the fetus and the pregnant person. On a scale from 0 (“no risk”) to 10 (“very high risk”), the average perceived risk was 8.01 (SD = 2.06) for the fetus and 6.68 (SD = 2.73) for the pregnant person. While responses showed a broad distribution, most ratings clustered at the upper end of the scale, indicating a generally high level of perceived risk (see Figure 2).

In comparison, the perceived risk of alcohol use during pregnancy was rated even higher. Alcohol was rated with a mean risk of M = 9.21 (SD = 1.34) for the fetus and M = 7.14 (SD = 2.32) for the pregnant person (see Figure 2). Wilcoxon signed-rank tests confirmed that alcohol use during pregnancy was perceived as significantly riskier than cannabis use, both for the fetus (Z = −6.71, *p* < 0.001) and the pregnant person (Z = −2.34, *p* = 0.019).

#### 3.3.2. Predictors of Risk Perception

Perceived risk ratings were similar across professions. There were no significant differences between midwives and physicians in their assessment of fetal risk t(280.26) = −1.77, *p* = 0.078, d = 0.21 or maternal risk, t(281.90) = −0.50, *p* = 0.615, d = 0.06. According to Cohen’s classification, both effect sizes are considered small.

A weak positive correlation was found between self-rated knowledge and perceived risk for the fetus (ρ = 0.20, *p* = 0.016) and a slightly stronger correlation for the pregnant person (ρ = 0.26, *p* = 0.002). Older age was also associated with higher risk perception for both the fetus (ρ = 0.30, *p* = < 0.001) and the pregnant person (ρ = 0.27, *p* = 0.001).

To examine the joint effect of knowledge and age, two multiple regression models were computed. For fetal risk perception, age was a significant predictor (β = 0.26, *p* = 0.005), whereas knowledge was not (β = 0.11, *p* = 0.225). In contrast, both knowledge (β = 0.20, *p* = 0.026) and age (β = 0.20, *p* = 0.030) significantly predicted maternal risk perception.

### 3.4. Communication and Counseling Behavior Regarding Cannabis Use

#### 3.4.1. Frequency of Screening and Counseling on Cannabis Use

Table 3 displays the reported frequency of screening and counseling practices regarding cannabis use during pregnancy among midwives. In this study, screening refers to verbally asking pregnant women about their substance use during routine care. While 39.9% reported routinely asking about cannabis use, 21.0% stated that they never did so. Counseling was reported less frequently than screening, with only 21.7% always providing information about cannabis-related health risks, and 35.0% indicating that they never did so.

To contextualize these findings, screening behavior for alcohol use was included as a reference. A Wilcoxon signed-rank test showed that midwives screened significantly more often for alcohol use (M = 3.24, SD = 0.83) than for cannabis use (M = 2.77, SD = 1.19), Z = −4.65, *p* < 0.001.

#### 3.4.2. Influence of Profession

Group differences in communication behavior were examined by comparing midwives and physicians regarding the frequency with which they addressed cannabis use during pregnancy. On a 4-point scale (1 = never, 4 = always), midwives reported significantly higher mean scores for counseling frequency (M = 2.77, SD = 1.19) than physicians (M = 2.44, SD = 1.06), indicating that midwives were more likely to provide information about cannabis-related health risks, χ^2^ (3) = 14.40, *p* = 0.002.

In terms of counseling behavior, midwives were more likely than physicians to report that they never provided information on cannabis-related risks (35.0% vs. 19.9%). The overall distribution of counseling frequencies also differed significantly between groups, χ^2^ (3) = 9.58, *p* = 0.023 (see Figure 3).

#### 3.4.3. Influence of Age and Knowledge Level on Screening/Counseling Behavior

Spearman’s correlation revealed that self-rated knowledge about cannabis use was positively associated with both screening frequency (ρ = 0.25, *p* = 0.003) and counseling frequency (ρ = 0.43, *p* < 0.001), indicating more frequent communication among those with higher knowledge. Age showed a small positive correlation with counseling frequency (ρ = 0.19, *p* = 0.030), and no significant association with screening (ρ = 0.08, *p* = 0.369).

A multiple linear regression analysis was performed to examine the independent contribution of knowledge and age. The model predicting counseling frequency was significant, F (2132) = 14.93, *p* < 0.001, R^2^_adj = 0.172. Only knowledge emerged as a significant predictor (β = 0.42, *p* < 0.001), while age was not (β = 0.03, *p* = 0.705).

## 4. Discussion

This study explored midwives’ knowledge, risk perception, and communication practices regarding cannabis use during pregnancy in Germany, with a particular focus on the influence of training, age, and professional background. Comparisons with physicians provided additional context for interpreting the findings across professional groups. The findings offer valuable insights into midwives’ preparedness to address prenatal cannabis use—an issue gaining relevance due to increasing social acceptance and the legalization of recreational cannabis in Germany in 2024.

### 4.1. Summary of Key Findings

Midwives, on average, rated their knowledge about cannabis-related risks during pregnancy as moderate and notably lower than their knowledge of alcohol-related risks. While the majority indicated that the topic had been addressed during their initial training, only a minority had completed relevant continuing education. Importantly, attendance in such programs was significantly associated with higher self-rated knowledge. In addition, older age was also linked to greater perceived knowledge.

Risk perception was generally high, especially regarding fetal outcomes, and was positively associated with both age and self-rated knowledge. However, cannabis was perceived as less harmful than alcohol, which may influence prioritization in counseling.

Despite recognizing the potential harms, fewer than half of the midwives consistently addressed cannabis use in consultations, and only a fifth regularly provided risk counseling. Compared to alcohol, cannabis was discussed and counseled on significantly less frequently. Knowledge emerged as a significant predictor for communication behavior, whereas age showed no such effect.

### 4.2. Integration with Existing Literature

#### 4.2.1. Knowledge

The findings of this study are consistent with previous research indicating that cannabis use during pregnancy is insufficiently covered in professional education and clinical practice [30,31]. Similar to international studies, the topic appears to be addressed during training primarily in the context of general substance use without cannabis-specific content [29]. This may explain why many participants reported having encountered the issue during their initial training, yet this exposure did not translate into higher self-rated knowledge. This lack of focus may contribute to the lower knowledge ratings observed for cannabis compared to alcohol, despite growing prevalence among reproductive-age individuals.

The study highlights the importance of continuing education: only continuing education, not the inclusion of substance use topics during initial training, was associated with increased perceived knowledge. This supports previous findings that not only the presence, but the depth and quality of training play a crucial role in knowledge acquisition [29,31].

Furthermore, older midwives tended to rate their knowledge more highly, which may reflect greater cumulative professional experience, a higher likelihood of having attended further training sessions, or generational differences in self-perception and attitudes toward substance use.

#### 4.2.2. Risk Perception

Overall, risk perception was high—particularly concerning potential harm to the fetus. This is consistent with previous research suggesting that healthcare professionals generally perceive prenatal cannabis use as risky but may lack the detailed knowledge to articulate specific risks confidently [30].

Notably, cannabis was perceived as less harmful than alcohol, which may reflect the well-established evidence base and public awareness regarding alcohol-related risks in pregnancy, in contrast to the comparatively limited research on cannabis [29,30,31].

Risk perception correlated positively with both knowledge and age. Regression analyses confirmed age as a predictor for fetal risk perception and both age and knowledge as predictors for maternal risk perception. These results echo previous work emphasizing the influence of personal experience and professional confidence in shaping risk assessment [29,30].

#### 4.2.3. Communication Practices

Midwives in this study addressed cannabis use during pregnancy relatively infrequently. Only 40% of midwives routinely asked about cannabis use, and just 22% always provided counseling. Alcohol use was addressed more frequently, which is consistent with previous findings suggesting that the presence of established guidelines and greater clinical certainty facilitates professional communication [30,31].

This mirrors previous reports of professionals avoiding the topic due to uncertainty about evidence-based recommendations or fear of damaging the therapeutic relationship [29,30,31]. Inconsistent communication not only limits the effectiveness of care but may also reduce opportunities for disclosure by pregnant individuals.

Knowledge was the most consistent predictor of communication frequency, underlining the essential role of continuing education and the need to strengthen substance-related content in professional training. Adequate knowledge appears crucial to fostering both the competence and confidence required to initiate open, nonjudgmental conversations about cannabis use during pregnancy.

Although age was associated with higher perceived knowledge and risk perception, it did not influence communication frequency. This suggests that factual knowledge and targeted education may be more influential in shaping communication practices than professional experience alone.

### 4.3. Strengths and Limitations

A major strength of this study is its focus on midwives in a German-speaking context—a group largely underrepresented in previous research on prenatal cannabis use. Most existing studies have been conducted in countries such as the United States, where maternity care structures differ significantly. Furthermore, prior research has often focused broadly on various health professionals, rather than specifically on midwives. The sample included professionals with diverse backgrounds and levels of experience, enhancing the generalizability of the findings.

However, some limitations must be acknowledged. First, the study relied on self-reported data, which may be affected by social desirability or response biases. Moreover, the assessment of knowledge was based on subjective self-evaluation rather than objective testing, which could lead to over- or underestimation of actual knowledge levels.

As a secondary data analysis, the original survey was not specifically designed to explore cannabis use in depth. Consequently, some relevant aspects may not have been fully captured, such as the extent and specificity of questions regarding cannabis use, as well as detailed information on the nature, duration, and quality of counseling practices. Moreover, certain items, such as those related to stigma, were only posed in the context of alcohol use, limiting the ability to investigate these aspects in relation to cannabis. Additionally, as the survey focused specifically on prenatal cannabis exposure, no data on cannabis use during lactation were collected.

The cross-sectional design further limits the ability to draw causal inferences. While the data reveal associations between knowledge, risk perception, and communication practices, it remains unclear in which direction these relationships operate.

Given the exploratory nature of the study and the multiple comparisons conducted without adjustment, findings should be interpreted as descriptive rather than confirmatory. Further hypothesis-driven research with larger, representative samples is needed to validate these observations.

The use of a convenience sampling strategy, including snowball recruitment, also limits representativeness. However, the high proportion of midwives working in non- hospital settings is not entirely unrepresentative, as approximately 70% of midwives in Germany work on a freelance basis [36]. These midwives often provide longer-term, trust-based care, potentially increasing their influence on clients’ health-related behaviors, including substance use.

The sample’s average age of approximately 38.8 years aligns reasonably well with the national average of 41 years [37]. Although the participation of a male midwife was not expected given the small sample size and the very low proportion of male midwives in Germany (0.08%) [38], one male participant was included in the sample. A randomized or stratified sampling strategy would have enhanced the representativeness of the sample and reduced potential sampling bias.

Moreover, random sampling would also have helped balance differences between and within groups—such as variations in age, professional experience, or work setting. This heterogeneity may have influenced group comparisons and limited the ability to draw specific conclusions. While multivariate analyses are commonly used to statistically control for such confounding variables, they were applied only to a limited extent in this study. More frequent use of multivariate methods could have reduced potential bias and enabled a more accurate interpretation of group differences, independent of structural differences between or within the samples.

The comparison group of physicians introduced additional complexity. The survey was distributed to gynecologists, so most participating physicians likely specialized in this field. However, participation from physicians of other specialties cannot be entirely ruled out, as no formal verification of specialty was conducted. In addition, the physician sample was generally older and more heterogeneous in terms of background, which may have affected comparability and should be considered in future research.

Participation was voluntary, which may have led to self-selection bias. It is likely that midwives with a particular interest or expertise in substance use were more inclined to participate. These individuals may be more informed and more proactive in addressing cannabis use than the broader midwifery population.

Finally, the analysis of communication practices focused on frequency rather than content, depth, or barriers to implementation. As such, important qualitative aspects of the counseling process remain unexplored.

### 4.4. Implications

This study supports the hypothesis that there is a clear need for targeted continuing education on prenatal cannabis use, with a focus on improving both knowledge and communication practices. Given the shifting legal and cultural landscape, maternal healthcare providers must be equipped to address cannabis use in a clear, empathetic, and evidence-based manner. Midwives should be equipped with up-to-date, evidence-based information and guided through practical approaches for addressing cannabis use in a nonjudgmental, supportive manner. Professional guidelines and standardized tools for communication, such as structured screening questions and counseling protocols, could enhance consistency and reduce uncertainty. Training should include not only factual knowledge but also communication techniques such as motivational interviewing.

Importantly, given the constraints of existing curricula, the integration of cannabis-specific content into midwifery education does not necessarily require the development of entirely new training modules. Instead, synergies with existing content—such as instruction on alcohol-related risks—could be leveraged to address shared principles of risk communication and substance use in pregnancy more broadly.

Further research is needed to explore the effectiveness of different educational formats in enhancing professionals’ competencies regarding cannabis use in pregnancy. Longitudinal studies could help establish causal pathways between knowledge acquisition, changes in risk perception, and improvements in communication practices. Additionally, qualitative studies may provide deeper insight into midwives’ subjective experiences and barriers in discussing cannabis use.

Future studies should also investigate cannabis use during lactation, as midwives play a central role in postpartum care and counseling.

The findings underscore the importance of including cannabis-specific content in both undergraduate curricula and post-graduate training for midwives. Given the changing legal landscape and growing prevalence of cannabis use, maternal health education must evolve accordingly to ensure that midwives are adequately prepared to provide comprehensive and empathetic care.

## 5. Conclusions

While most midwives had encountered the topic of substance use during their initial training, only continuing education and older age were associated with higher self-rated knowledge about cannabis-related risks. Knowledge emerged as an important factor influencing both risk perception and communication behavior. To strengthen midwives’ preparedness in addressing cannabis use during pregnancy, integrating more cannabis-specific content into professional education and expanding targeted continuing education opportunities may be helpful. In light of increasing legalization and social acceptance, clear, evidence-based, and non-stigmatizing communication strategies remain essential to support maternal and fetal health.

## Figures and Tables

**Figure 1 healthcare-13-01228-f001:**
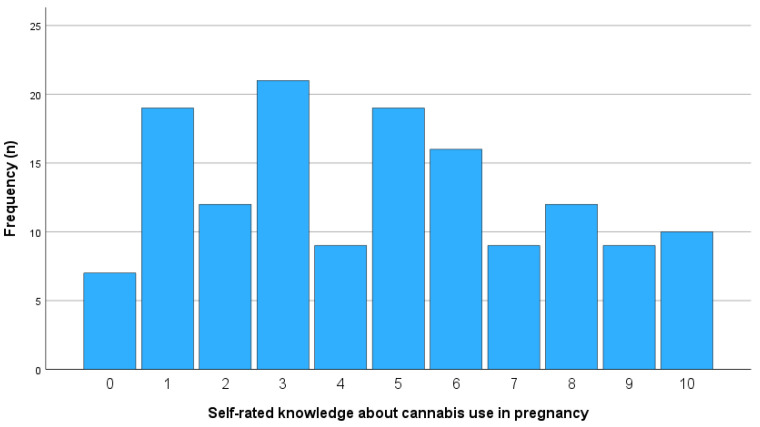
Self-rated knowledge of midwives about cannabis use during pregnancy (0–10 scale).

**Figure 2 healthcare-13-01228-f002:**
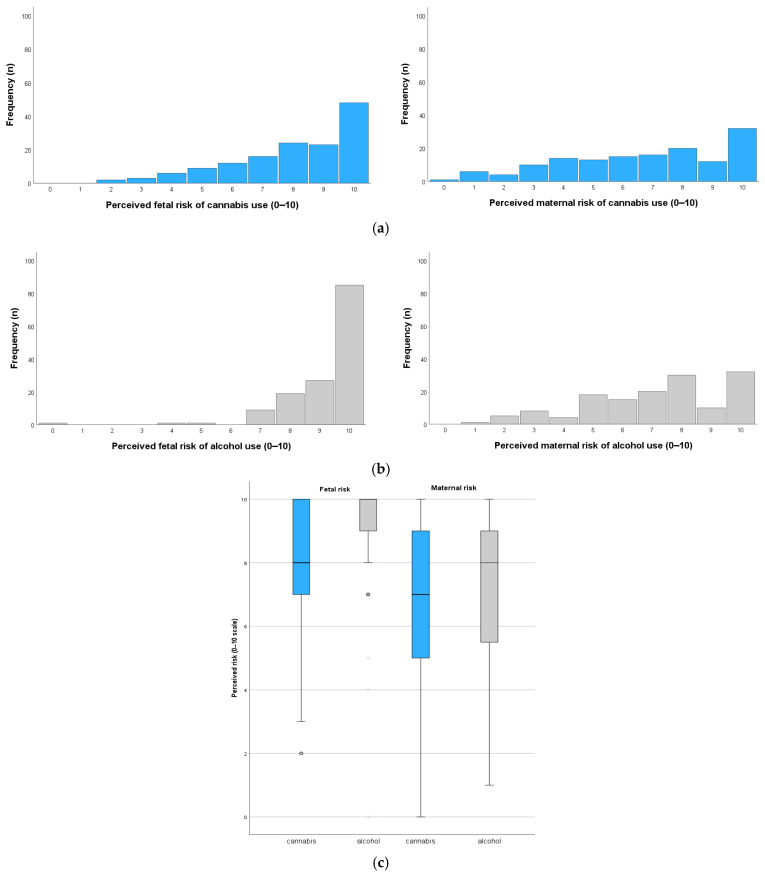
Perceived risk of prenatal cannabis and alcohol use among midwives. (**a**) Distribution of perceived fetal and maternal risks of cannabis use during pregnancy. (**b**) Distribution of perceived fetal and maternal risks of alcohol use during pregnancy. (**c**) Boxplot comparison of perceived fetal and maternal risks for cannabis versus alcohol use.

**Figure 3 healthcare-13-01228-f003:**
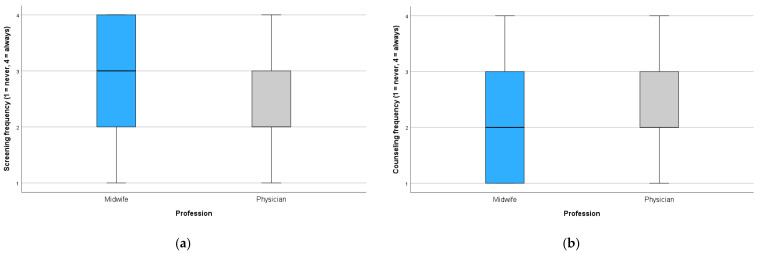
Frequency of (**a**) screening and (**b**) counseling for cannabis use during pregnancy, stratified by profession (1 = never, 2 = rarely, 3 = mostly, 4 = always).

**Table 1 healthcare-13-01228-t001:** Overview of variables included in the statistical analyses.

Variable	Type/Scale	Coding/Range	Statistical Tests Applied
Profession	Categorical (nominal)	Midwife, Physician	χ^2^-test, *t*-test
Age	Continuous (metric)	Years	Spearman correlation, regression
Years of professional experience	Continuous (metric)	Years	Not used (strongly correlated with age: ρ = 0.94, *p* < 0.001)
Addiction/substance use was part of the professional training	Categorical (nominal)	Yes, No	χ^2^-test, *t*-test, regression
Attended further training on addiction/substance	Categorical (nominal)	Yes, No	χ^2^-test, *t*-test, regression
Self-rated knowledge	Ordinal/ Interval	0 = no knowledge to 10 = very high knowledge	*t*-test, Spearman correlation, regression
Risk perception (child)	Ordinal/ Interval	0 = no risk to 10 = very high risk	*t*-test, Wilcoxon signed-rank, Spearman correlation, regression
Risk perception (pregnant person)	Ordinal/ Interval	0 = no risk to 10 = very high risk	*t*-test, Wilcoxon signed-rank, Spearman correlation, regression
Screening frequency	Ordinal (4-point)	1 = never, 4 = always	χ^2^-test, Wilcoxon signed-rank test, Spearman correlation, regression
Counseling frequency	Ordinal (4-point)	1 = never, 4 = always	χ^2^-test, Spearman correlation, regression

**Table 2 healthcare-13-01228-t002:** Sociodemographic and professional characteristics of midwives and physicians.

Characteristic	Midwives (n = 143)	Physicians (n = 141)	Total (N = 284)
**Age (years), M (SD) ^1^**	38.8 (11.6)	45.9 (10.3)	42.3 (11.4)
**Biological sex, n (%)**			
Male	1 (0.7%)	12 (8.5%)	13 (4.6%)
Female	142 (99.3%)	129 (91.5%)	271 (95.4%)
Divers	–	–	–
No statement	7 (4.9%)	3 (2.1%)	10 (3.5%)
**Social gender, n (%)**			
Cis	136 (95.1%)	138 (97.9%)	274 (96.5%)
Trans	2 (1.4%)		2 (0.7%)
No statement	5 (3.5%)	3 (2.1%)	8 (2.8%)
**Personal cannabis consumption**			
Yes	9 (6.3%)	4 (2.8%)	13 (4.6%)
No	133 (93.0%)	137 (97.2%)	270 (95.1%)
No statement	1 (0.7%)	–	1 (0.4%)
**Work experience (years), M (SD)**	14.3 (11.8)	18.5 (10.2)	16.4 (11.1)
**Work context**			
University hospital	10 (7%)	23 (16.3%)	33 (11.6%)
Hospital	43 (30.1%)	20 (14.2%)	63 (22.2%)
Doctor’s office	5 (3.5%)	92 (65.2%)	97 (34.2%)
Midwifery practice/birth center/self-employed	78 (54.5%)	–	78 (27.5%)
Advice center	1 (0.7%)	2 (1.4%)	3 (1.1%)
Other	6 (4.2%)	4 (2.8%)	10 (3.5%)
**Addiction/substance use was part of the professional training**			
Yes	112 (78.3%)	92 (65.2%)	204 (71.8%)
No	31 (21.7%)	49 (34.8%)	80 (28.2%)
**Attended further training on addiction/substance**			
Yes	51 (35.7%)	49 (34.8%)	100 (35.2%)
No	92 (64.3%)	92 (65.2%)	184 (64.8%)

^1^ Specification of age was optional; total sample size for this variable was therefore smaller (N = 271).

**Table 3 healthcare-13-01228-t003:** Frequency of screening and counseling on cannabis use during pregnancy.

Frequency	Screening (%)	Counseling (%)
Never (1)	30 (21.0%)	50 (35.0%)
Rarely (2)	30 (21.0%)	38 (26.6%)
Mostly (3)	26 (18.2%)	24 (16.8%)
Always (4)	57 (39.9%)	31 (21.7%)
Mean (SD)	2.77 (1.19)	2.08 (1.15)

## Data Availability

The data are not publicly available due to privacy and ethical restrictions. Anonymized datasets may be made available by the corresponding author upon reasonable request and subject to approval by the responsible ethics committee.

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
