# Peer review of "Pregnancy Care in Times of Cannabis Legalization: Self-Rated Knowledge, Risk Perception and Communication Practices of Midwives in Germany"

_healthcare, 2025, doi:10.3390/healthcare13111228_

Round 1
Reviewer 1 Report
Comments and Suggestions for Authors
I have read this paper with great interest, as the German scenario is rather an example, since the shifts on legalisation occur throughout the (Western) world. I value the effort, although is remains quite difficult to provide accurate information on a setting of still the relevant uncertainties related to cannabis exposure during pregnancy, while I do agree that the final outcome is likely modulated by a lot of co-factors involved (as also mentioned by the authors).
I’m surprise not to find any reflection on lactation related exposure, especially when the focus is on midwives. Have you collected such data, and only focus on pregnancy related aspects in this paper, or it this ‘just’ a shortage, that is likely useful to be added to the paper.
We need some more information on the pregnancy related training, suggested in the 2.1 section of the paper. How do the timelines look like, were the questionnaires collected after the training, before, or both ? and were respondents attendees of the training (motivation related).
Line 217: non-clinical, perhaps non-hospital ?
Do you ‘know’ to what extent the ‘sample of midwives’ reflects the German population of midwives on eg age, work place or other ?
How should I understand ‘screening’, is this explicit asking the subject on cannabis use, or is a lab test (saliva, urine or blood).
4.2.1. likely ‘perceived’ knowledge ?
Reviewer 2 Report
Comments and Suggestions for Authors
Please find the suggestions in the attached document.

Reviewer 3 Report
Comments and Suggestions for Authors
The paper addresses a specific and timely research gap in maternal healthcare by exploring how midwives in Germany understand and manage cannabis use during pregnancy in the context of its legalization in 2024. The paper is one of the first to examine maternal healthcare in Germany post-legalization of recreational cannabis. Based on the article the authors could improve the methodology and consider additional controls in the following ways:
- A randomized or stratified sampling approach could reduce bias and better represent the midwifery population.
- The physician comparison group was broad and included various specialties. Future studies should use a more homogeneous control group to ensure meaningful comparisons.
- While a pilot test was done, the paper does not mention psychometric validation of the questionnaire.
Implementing these improvements and controls would enhance the study's rigor, validity, and policy implications.
